# DiagramDiff: A Diagram Reconstruction and Recognition Method to Enhance Large Language Models' Diagram Understanding

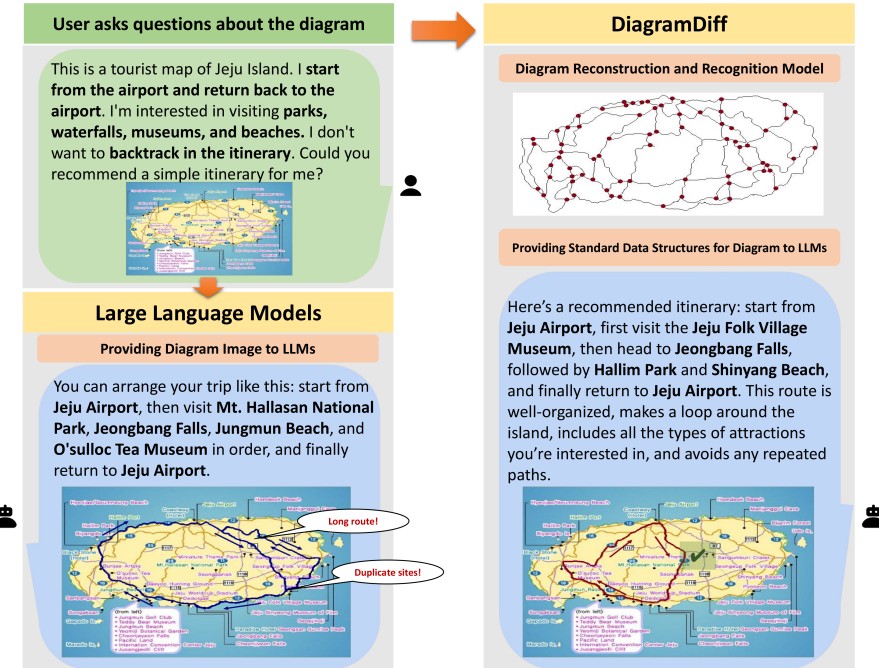

Figure 1: The user provides the offline map image and request to both LLMs and DiagramDiff. Our method, through diagram reconstruction and recognition, provides standardized diagram data to the LLMs. As a result, LLMs show significant back-and-forth route issues and offer longer incorrect routes, while our method provides a more reasonable route that meets the user's requirements.

## Abstract

Diagrams are widely used in daily life. However, offline diagrams usually exist as images, lacking structured data representation, which limits their reusability and editability. Current research mainly supports basic query tasks for online diagrams and does not meet semantic understanding or interaction needs for complex offline diagrams. Although large language models (LLMs) possess strong reasoning and knowledge integration abilities, their performance on offline diagrams is unsatisfactory due to difficulty in accurately understanding their structure and content. To address these issues, we propose DiagramDiff, a framework with a high-precision diagram reconstruction model and an instance-level diagram element recognition model. The framework converts offline diagrams into standardized structures, enabling LLMs to move from limited understanding to intelligent assistants capable of semantic reasoning, logical validation, and efficient editing. We also built a dataset with diagrams and corresponding Q&A and editing tasks. Experiments show DiagramDiff achieves state-of-the-art performance, significantly enhancing LLMs' diagram understanding and interaction.

# 1 INTRODUCTION

Diagrams serve as essential tools for conveying information in scientific research education, and software development (Ma'ayan et al., 2020), providing intuitive representations of complex models' structures and logic. However, offline diagrams, often existing in image form, lack structured data representations, severely limiting their reusability and editability. Manual diagram redrawing is typically required for modifications, a process that is both operation-intensive and error-prone. Furthermore, current interactive methods for offline diagrams are limited. Existing studies focus primarily on simple Q&A for online diagrams and lack support for semantic understanding and interactive operations for complex offline diagrams. LLMs excel at reasoning and knowledge integration tasks, including image semantic understanding, logical query resolution, and structured data processing. However, when applied to complex diagram images, LLMs like GPT-4o(OpenAI, 2023) struggle with precise diagram understanding, leading to reduced accuracy in Q&A and editing tasks. Existing studies have yet to integrate LLMs effectively into offline diagram interactions. Current methods primarily rely on pre-constructed knowledge bases for diagram Q&A, using natural language interfaces for information retrieval or content highlighting.Therefore, enhancing the ability of LLMs to understand and interact with offline diagrams is of significant importance. By constructing standardized and structured diagram data, LLMs can be transformed from tools limited to answering simple questions into intelligent assistants capable of supporting complex semantic reasoning, logical validation, and efficient content editing. Achieving this objective requires high-precision stroke reconstruction and recognition methods to enable instance-level stroke identification and the construction of standardized diagram data structures. Existing stroke reconstruction methods for diagram images face significant limitations. Methods like pixel search and template matching are primarily designed for character recognition and cannot handle the complexities of diagram strokes. Region-based segmentation and instance segmentation methods struggle with challenges such as stroke breakage and fuzzy intersections, resulting in reduced accuracy in semantic understanding and structural analysis. Current offline diagram recognition methods are restricted to region-level identification and fail to achieve the precision required for instance-level stroke recognition. Moreover, reconstructed strokes often exhibit deviations in attributes features, such as inaccurate classifications, with no existing methods addressing these issues effectively.

To address these challenges, we propose DiagramDiff, a novel framework for offline diagram reconstruction and recognition that enhances LLMs' diagram comprehension capabilities, bridging the gap between offline diagrams and LLMs. As shown in Figure 1, when users require Q&A and editing on complex diagram images, existing LLMs are unable to accurately comprehend and provide the necessary services. The main contributions are summarized as follows:

- We propose a novel stroke reconstruction method for offline diagrams, enhancing their reusability and achieving state-of-the-art reconstruction accuracy.
- We introduce a novel diagram recognition method that integrates diffusion model into the GTN framework to address stroke attribute inaccuracies and generate standardized data, enabling LLMs to provide effective intelligent services for offline diagrams.
- We propose DiagramQAE, the first offline diagram Q&A and editing dataset to validate the effectiveness of our method. User experiments show that DiagramDiff significantly enhances LLMs' understanding of diagrams.

# 2 RELATED WORK

## 2.1 OFFLINE STROKE RECONSTRUCTION METHODS

Researchers have developed various offline stroke reconstruction techniques. Path-based methods (Song et al., 2022b) assess connections by exploring possible trajectories, necessitating assumptions about line positions and shapes, thus restricting their use to standardized diagrams. Semantic segmentation (Song et al., 2022a) can outline line regions but often produce broken lines and struggle with intersections common in diagrams. Similarly, edge detection (Canny, 1986) can capture curves but fail to reliably reconstruct intersecting lines. In handwriting analysis, stroke reconstruction aims to derive continuous stroke data, such as coordinate sequences, from static images to facilitate tasks like character recognition. Pixel-based search approaches (Chan, 2020) enhance formula recognition

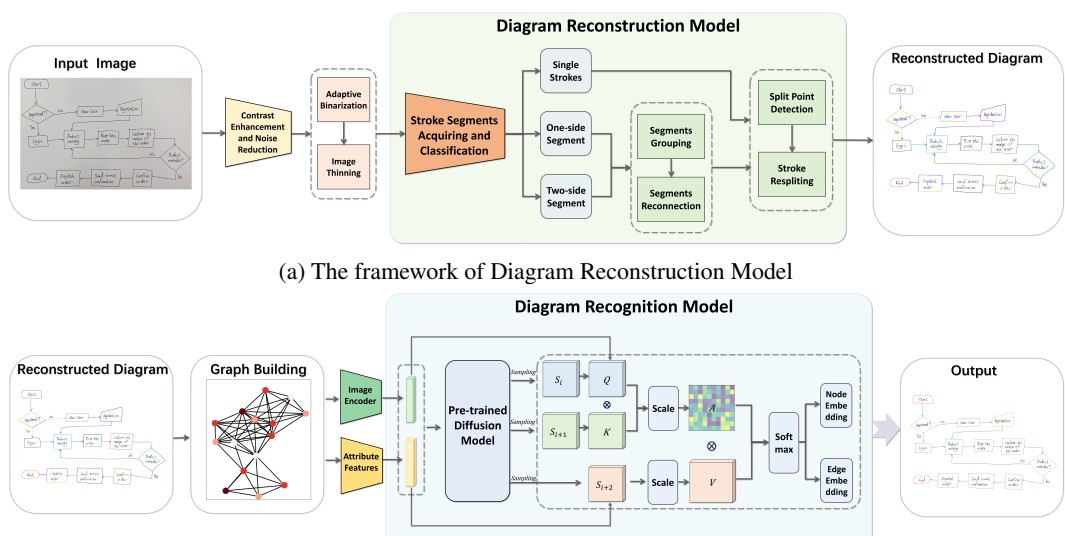

(a) The framework of Diagram Reconstruction Model

(b) The framework of Diagram Recognition Model

Figure 2: (a)The Diagram Reconstruction Model converts offline diagrams into online strokes for stroke-level recognition, employing endpoint-based segment searching, classification, grouping, and reconnection, followed by split point detection to distinguish separate elements. (b)The Diagram Recognition Model integrates the Diffusion Model with the GTN and employs a multi-sample attention mechanism to address attribute bias in offline diagram instance recognition.

accuracy by utilizing stroke information, while template matching methods (Li et al., 2023; Wang et al., 2022) and pen trajectory prediction (Mohamed Moussa et al., 2023) improve stroke extraction for specific languages. Nonetheless, current methods primarily address text-level strokes, lacking the capability to incorporate diagram semantics, templates, or establish connection relationships necessary for comprehensive diagram editing.

## 2.2 DIAGRAM RECOGNITION ALGORITHMS

Offline diagram recognition methods recognize diagrams in image form.such as Faster R-CNN (Montellano et al., 2022)(Julca-Aguilar & Hirata, 2018) and Arrow R-CNN (Schäfer et al., 2021), can identify elements in flowcharts but cannot achieve stroke-level instance segmentation, their accuracy drops significantly when handling complex diagrams with many nodes or overlapping connections (Song et al., 2023). Online diagram recognition methods can support stroke-level recognition, such as Instance GNN (Yun et al., 2022), DyGAT (Yang et al., 2023) and SpaceGTN (Hu et al., 2024), but these online methods cannot be applied to offline diagram recognition.

## 3 METHOD

### 3.1 OFFLINE DIAGRAM RECONSTRUCTION

In this part, we extract strokes from diagram images (offline diagrams), encompassing both node strokes, connector lines and handwritten text, which collectively describe the structural information of the diagram. Existing works have studied stroke extraction from handwritten characters(Chan, 2020; Mohamed Moussa et al., 2023). They usually reconstruct the handwriting based on template matching or predicting the writing direction, but may require domain prior knowledge or training on different data. As illustrated in Figure 2a, we proposed a more general language-independent method that processes diagrams and handwritten text to reconstruct the diagram structure and stroke categories. To achieve this, we implement decomposition and recombination by segmenting strokes and selectively connecting segments to obtain complete strokes. and develop an innovative split point detection module that distinguishes between separate instances of identical strokes, thereby

enhancing the accuracy of stroke classification and diagram recognition. Images with dense and fuzzy crossings are not specially addressed in our method as they tend to be less readable and may need tailored visualization or interactive tools for effective analysis.

### 3.1.1 SEGMENTS CLASSIFICATION

We utilize the Guo-Hall algorithm(Guo & Hall, 1989) to refine the image and partition it into stroke segments, which defined as continuous stroke sequences without bifurcations. Strokes are binarized for simplicity, since diagrams typically show clear contrast, which also supports processing of images with colors and additional elements. Specifically, we identify joint points within the diagram by categorizing pixels into background, stroke points, and joint points. Background points refer to points in areas without strokes and joint points are potential intersection points where multiple strokes may converge, but not necessarily indicate actual stroke intersections, implying that stroke segments might not always be maximally delineated. We first label stroke points and then identify joint points by examining whether a stroke pixel's neighboring regions separate the background into three or more distinct areas. This can be calculated by 8-connected component of the background:

$$C(p) = \sum_{S \subseteq B(p)} \begin{cases} 1, & \text{if } S \text{ is 8-connected and maximal,} \\ 0, & \text{otherwise.} \end{cases} \tag{1}$$

where $B(p)$ is the set of background pixels, $p$ denotes a stroke point and $C(p)$ counts background areas around $p$. The maximal condition prevents background areas from connecting. To suppress noise, a pixel-response value $R(p)$ is integrated, yielding the joint-point probability

$$P_j(p) = \sigma\Big(\eta\, C(p) + (1 - \eta)\, \hat{R}(p)\Big), \tag{2}$$

where $\hat{R}(p) = R(p)/\max_{q \in \mathcal{S}} R(q)$, $\eta \in [0, 1]$ balances the two terms, and $\sigma$ is the sigmoid. A pixel is marked as a joint point when $P_j(p) > \tau_j$. Stroke segments are classified into three categories: (1) segments with both ends as joint points, (2) segments with one end as a joint point and the other end open (not connected to other segments), and (3) segments with both ends open. For (3) each stroke segment is considered complete and isolated, and the remaining stroke segments should be combined to form complete strokes, addressing the inevitable intersections present in diagrams.

### 3.1.2 STROKE RECONSTRUCTION

We merge adjacent joint points to form groups of stroke segments, where two or more segments may be contiguous near a joint point. Since strokes may intersect in X- or T-shaped forms, we assume that a stroke preserves its continuity through the intersection, while different strokes remain separate, in line with human sketch perception. By iterating through pairs within each group, we calculate the degree of connectivity between stroke segments. The connectivity $C_{i,j}$ between two stroke segments $S_i$ and $S_j$ is evaluated by considering their spatial proximity, angular alignment, and curvature similarity. It is defined by the following equation:

$$C_{i,j} = \alpha e^{-\lambda d_{\min}} + \beta \cos(\Delta\theta) + \gamma e^{-\mu|k_i - k_j|} \tag{3}$$

where $d_{\min}$ is the minimum Euclidean distance between the endpoints of $S_i$ and $S_j$. $\Delta\theta$ is the absolute difference in orientation angles of $S_i$ and $S_j$. $k_i$ and $k_j$ represent the average curvatures of $S_i$ and $S_j$, respectively. $\alpha, \beta, \gamma$ are weighting coefficients that balance the influence of each factor. $\lambda$ and $\mu$ are scaling parameters controlling the sensitivity to distance and curvature differences.This formula integrates the closeness of the stroke segments, their directional alignment, and the similarity in their curvature to quantify the likelihood that $S_i$ and $S_j$ are connected within the diagram. Since every primitive segment has only two endpoints and each endpoint must be used at most once in the iteration, selecting connections is formulated as a *matching* problem:

$$\mathcal{M}^\star = \arg\max_{\mathcal{M} \subseteq \mathcal{E}} \sum_{(i,j) \in \mathcal{M}} C_{i,j} \quad \text{s.t. } \deg_{\mathcal{M}}(i) \leq 1, \ \forall\, i. \tag{4}$$

Distinct from stroke extraction methods tailored for character recognition, our method necessitates the categorization of each stroke to differentiate between various diagram elements. A challenge arises as traditional stroke extraction methods might erroneously merge distinct strokes into a single

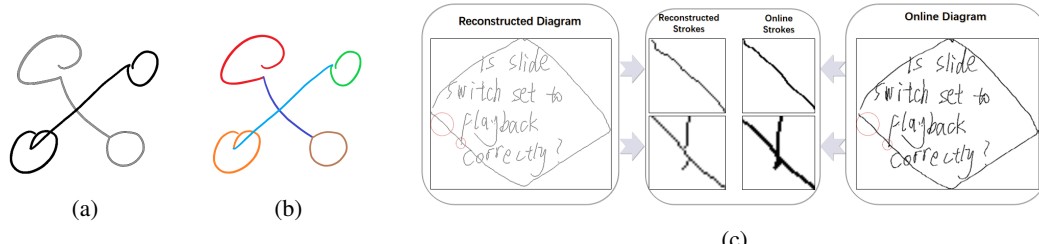

(a)    (b)

(c)

Figure 3: (a) shows two single strokes each representing a line linking two circles, (b) shows the expected stroke splitting result, and (c) shows an example of the online and reconstructed offline diagrams with a magnified comparison of regions.

entity. Consider the example shown in Figure 3aFigure 3b, a diagram where a user draws two circles and a connecting line in one continuous stroke might be incorrectly interpreted as a single stroke, whereas it should be recognized as three separate strokes. To address this, we implement split-point detection for all extracted strokes, identifying points at local sharp angles as potential split locations. We perform corner detection on strokes by applying a sliding window to evaluate the angles between the central point and its endpoints within the window. Different from a fixed angular threshold, a length-aware window threshold is used:

$$\theta_{\text{thr}}(\ell) = \theta_0 + \rho\, e^{-\ell/\ell_0}, \tag{5}$$

where $\ell$ is the current window length, $\theta_0$ the minimum angle, and $\rho$ the decay rate. When the observed angle $\theta < \theta_{\text{thr}}(\ell)$, the window centre is a candidate split point. Curvature is then estimated with the three-point circle method

$$\kappa = \frac{2\left|x_1\Delta y_{23} + x_2\Delta y_{31} + x_3\Delta y_{12}\right|}{\sqrt{\Delta x_{23}^2 + \Delta y_{23}^2}\,\sqrt{\Delta x_{13}^2 + \Delta y_{13}^2}\,\sqrt{\Delta x_{12}^2 + \Delta y_{12}^2}}, \tag{6}$$

where $\Delta x_{ij} = x_i - x_j$ and $\Delta y_{ij} = y_i - y_j$. To obtain a consistent split–merge decision, the candidates are optimised in a graph-cut framework. Let $e_s \in \{0, 1\}$ denote whether a candidate split point $s$ is cut (1) or kept (0), the global energy is

$$E(\mathbf{e}) = \sum_{s\in\mathcal{E}}\Big(\lambda_s\, e_s + \sum_{t\in\mathcal{N}(s)} \phi_{s,t}\left|e_s - e_t\right|\Big), \tag{7}$$

where $\lambda_s \propto \kappa_s$ penalises redundant cuts and $\phi_{s,t}$ enforces local consistency. The optimal configuration $\mathbf{e}^\star$ is obtained with $\alpha$–$\beta$ swap, producing globally optimised split points. The detailed algorithm process can be found in Appendix A.

## 3.2 DIAGRAM RECOGNITION MODEL

We propose a novel framework that integrates a Diffusion Model (Rombach et al., 2022) into a Graph Transformer Network (GTN) (Ying et al., 2021) to address attribute bias in offline diagram instance segmentation. As illustrated in Figure 3c, a key challenge is that stroke reconstruction introduces inaccuracies (e.g., jagged edges), leading to biased attributes such as curvature and thickness, which degrade the performance of existing methods. Conventional GTN-based approaches often rely on simple concatenation of stroke image features and attribute features. This naive fusion strategy struggles with biased attributes, as it fails to differentiate the primary role of reliable image features from the auxiliary role of potentially inaccurate attribute features. To overcome this limitation, our framework employs a novel feature fusion mechanism. The Diffusion Model treats image features as the core input, providing fine-grained structural detail, while incorporating attribute features as conditional inputs to guide the semantic generation process. Through its iterative denoising process, the Diffusion Model deeply fuses these two feature types, generating high-quality latent representations. This approach allows the robust image features to compensate for information loss caused by attribute bias, thereby effectively mitigating the impact of inaccurate attributes on recognition performance. Details on the graph building process are provided in Appendix B.

### 3.2.1 Feature Extraction

We propose a dual-channel framework to capture both geometric attributes and deep image features of strokes, scaling them for the pre-trained diffusion model. The first channel uses GCN (Kipf & Welling, 2016) to aggregate spatial features $F^a$, capturing geometric and contextual information, with a scaling layer for dimension alignment. The second channel extracts deep image features $F^i$ using a depthwise separable convolution (DWConv) module, incorporating a scaling layer to match the diffusion model's input dimensions.

### 3.2.2 Feature Enhancement with Diffusion Model

We propose a novel framework that integrates Diffusion Model into the Graph Transformer Network to enhance the performance of offline diagram instance segmentation. Specifically, during the sampling process, the intermediate denoising results generated by the diffusion model are utilized to provide additional feature representations for each stroke node, which complement the stroke attribute features and stroke image features. To reduce computational overhead, the DDIM sampling strategy (Song et al., 2020) is adopted. Meanwhile, considering the inherent randomness of the diffusion model's generated results, robustness is enhanced by performing three independent samplings. Through a single-step denoising process, three intermediate sampling results are extracted for each node $j$ in the graph. These feature representations are formally expressed as:

$$\{s_{1,j}, s_{2,j}, s_{3,j}\} \quad \text{for } j = 1, \ldots, n \tag{8}$$

where $s_{1,j}$, $s_{2,j}$, and $s_{3,j}$ represent the features generated by the diffusion model for the $j$-th node. These sampling results are subsequently used as the Query ($Q$), Key ($K$), and Value ($V$) inputs in the attention mechanism of the GTN, enabling the computation of attention weights and feature aggregation. The attention mechanism is formulated as:

$$O_j = \text{softmax}\left(\frac{s_{1,j}s_{2,j}^T}{\sqrt{d}}\right) \cdot s_{3,j} \oplus GCN(F_j^a) \oplus DWConv(F_j^i) \tag{9}$$

where $O_j$ represents the updated feature of the $j$-th node, $F_j^a$ denotes the attribute feature of the $j$-th node, and $F_j^i$ refers to the image feature of the $j$-th node.

### 3.3 Standardized Data Structure for Diagram

We propose a standardized data structure that categorizes reconstructed diagram elements into two classes: nodes and edges. Table 1 details the attributes of each class. We have converted offline diagrams, which were previously incomprehensible to LLMs, into a standard data structure that can be accurately comprehended by LLMs, through the Diagram Reconstruction Model and Diagram Recognition Model. Our method enhances the LLMs diagram comprehension ability by feeding diagram image and standardized diagram data, and providing the LLMs with the standardized diagram data structure we designed.

| Nodes | | | Edges | |
|---|---|---|---|---|
| **Attribute** | **Description** | | **Attribute** | **Description** |
| **ID** | Unique identifier | | **ID** | Unique identifier |
| **Text** | Node label or content | | **Text** | Edge label or content |
| **Type** | Node category | | **Type** | Edge category |
| **Size** | Node dimensions | | **Length** | Edge length |
| **Incoming Edges** | Incoming edge list | | **Direction** | Flow direction |
| **Outgoing Edges** | Outgoing edge list | | **Weight** | Connection strength |
| **Child Nodes** | Subordinate nodes | | **Start Points** | Starting points |
| **Parent Nodes** | Superior nodes | | **End Points** | Ending points |

Table 1: Detailed Node and Edge Attributes

## 4 DiagramQAE Dataset

We propose DiagramQAE, the first offline diagram Q&A and editing dataset designed to evaluate the effectiveness of our method in enhancing the question-answering and editing capabilities of

LLMs. This dataset comprises 100 manually created diagrams, including handwritten and digital diagrams, with three categories: flowcharts, mind maps, and state machine diagrams, containing a total of 3,186 symbols. Each diagram contains more than 10 symbols, and across the dataset, there are 20 different types of diagram symbols, highlighting the structural complexity of the data. The dataset was collaboratively created by 20 contributors. This includes stroke categories and their corresponding diagram elements, enabling a fine-grained analysis of handwriting patterns and diagram construction processes. Each diagram is paired with five carefully designed tasks, three for question answering and two for editing, resulting in a total of 500 diagram-related tasks. For each task, correct answers and corresponding correct editing results are provided, ensuring consistency and facilitating quantitative evaluation. As illustrated in Figure 4, these tasks are intentionally designed with varying levels of complexity, covering sophisticated diagram-based Q&A scenarios (such as reasoning and semantic understanding of complex diagrams.) and editing operations (such as modifying node connections, adjusting attribute information, or identifying errors in the diagram based on the order and correcting them). This comprehensive dataset not only supports the assessment of LLMs' abilities in understanding diagrams, but also provides a valuable resource for future research in diagram understanding, multimodal learning, and human-computer interaction. The categories of diagram elements included in the dataset can be found in Appendix C. The detailed contents of the dataset can be found in the supplementary materials.

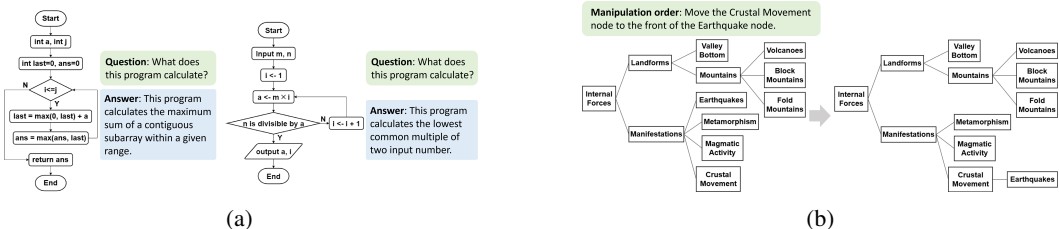

(a)                    (b)

Figure 4: (a) shows two Q&A examples and (b) shows an editing example.

# 5 EXPERIMENTS

## 5.1 DIAGRAM RECONSTRUCTION

### 5.1.1 EXPERIMENT DESIGN

We evaluated the model using the CASIA-OHFC and OHSD (Hu et al., 2024) datasets. We generated the test images from the online data. We leveraged and compared with methods of ( Liu et al. (2001), Chan (2020), Mohamed Moussa et al. (2023)), including the latest method in the field. We evaluate the performance of the model using intersection-over-union (IoU), stroke reconstruct rate (SRR) and Hausdorff distance (HD), defined as follows:

$$\text{IoU} = \frac{|P \cap T|}{|P \cup T|}, \quad \text{SRR} = \frac{|\{c(s) \geq 50\%, s \in S\}|}{|S|}, \quad \text{HD} = \max\{\sup_{p \in P} \inf_{t \in T} d(p,t), \sup_{t \in T} \inf_{p \in P} d(t,p)\} \quad (10)$$

where $t$ represents the pixel in the ground-truth image $T$, $p$ represents the pixel in the predicted image $P$, $s$ represents the stroke in strokes $S$, and $c(s)$ is the covered rate of $s$. HD quantifies the shape discrepancy between the ground truth and predicted images, with a smaller HD value indicating a more accurate reconstruction of the strokes.

| Method | CASIA-OHFC | | | OHSD | | |
|---|---|---|---|---|---|---|
| | IoU(%) | SRR(%) | HD | IoU(%) | SRR(%) | HD |
| Liu et al. | 36.0 | 78.3 | 4.62 | 30.9 | 74.7 | 4.75 |
| Chan et al. | 50.4 | 89.2 | 3.75 | 48 | 89.1 | 3.89 |
| Mouss et al. | 50.5 | 92.3 | 3.71 | 49.9 | 91.8 | 3.77 |
| **Ours** | **55.6** | **96.5** | **3.61** | **53.9** | **96.2** | **3.69** |

Table 2: Comparison of state-of-the-art methods for diagram reconstruction.

| Method | Editting Task | | Q&A Task | |
|---|---|---|---|---|
| | Original | **Ours** | Original | **Ours** |
| GPT 4o | 71% | **90%** | 77% | **92%** |
| Claude 3.7 | 50.5% | **61%** | 54% | **68.5%** |
| DeepSeek | 62% | **85%** | 69% | **80%** |
| GPT 4.5 | 69% | **86%** | 77.5% | **93%** |

Table 3: DiagramDiff performance on different LLMs

| Method | CASIA-OHFC | | OHSD | | FC_A | | FC_B | |
|---|---|---|---|---|---|---|---|---|
| | SCA | SCP | SCA | SCP | SCA | SCP | SCA | SCP |
| DETR | 49.74 | 51.45 | 46.73 | 45.26 | 46.32 | 52.19 | 66.49 | 63.09 |
| DeDETR | 50.56 | 58.16 | 55.49 | 60.45 | 54.11 | 54.98 | 62.75 | 65.87 |
| FasterRCNN | 69.18 | 71.28 | 65.37 | 67.41 | 69.42 | 67.38 | 72.56 | 76.36 |
| MaskRCNN | 69.17 | 67.53 | 74.11 | 69.53 | 75.34 | 72.13 | 76.25 | 70.41 |
| ORSAD | 85.24 | 81.99 | 83.29 | 79.91 | 92.99 | 90.12 | 94.89 | 93.46 |
| EGAT | 83.26 | 83.45 | 83.59 | 81.62 | 92.51 | 91.12 | 93.24 | 92.12 |
| InstGNN | 87.23 | 87.10 | 91.24 | 90.23 | 94.78 | 93.65 | 95.04 | 94.81 |
| SpaceGTN | 89.56 | 89.34 | 93.21 | 92.18 | 96.12 | 95.43 | 96.72 | 96.48 |
| **Ours** | **94.64** | **93.41** | **98.39** | **96.31** | **97.24** | **96.60** | **97.92** | **97.24** |

Table 4: Comparative experiments (%) with state-of-the-art offline and online recognition methods.

| Method | CASIA-OHFC | | | | OHSD | | | |
|---|---|---|---|---|---|---|---|---|
| | Original | | Reconstructed | | Original | | Reconstructed | |
| | SCA | SCP | SCA | SCP | SCA | SCP | SCA | SCP |
| ORSAD | 91.31 | 91.04 | 85.24 | 81.99 | 86.65 | 84.36 | 83.29 | 79.91 |
| EGAT | 92.76 | 92.01 | 83.26 | 83.45 | 92.82 | 90.46 | 83.59 | 79.91 |
| InstGNN | 95.81 | 95.42 | 87.23 | 87.10 | 96.89 | 95.44 | 91.24 | 90.23 |
| SpaceGTN | 98.13 | 97.93 | 89.56 | 89.34 | 99.78 | 99.32 | 93.21 | 92.18 |
| Ours (w/o FE) | - | - | 90.56 | 90.42 | - | - | 94.33 | 93.20 |
| **Ours (with FE)** | - | - | **94.64** | **93.41** | - | - | **98.39** | **96.31** |

Table 5: The performance (%) of online recognition methods on original and reconstructed diagrams.

## 5.1.2 RESULTS AND ANALYSIS

As shown in Table 2, our method achieved state-of-the-art performance in this field, realizing high-precision stroke reconstruction for diagrams.

## 5.2 DIAGRAM RECOGNITION

### 5.2.1 EXPERIMENT DESIGN

We conducted a comprehensive evaluation of the model using the FC_A (Awal et al., 2011), FC_B (Bresler et al., 2016), CASIA-OHFC, and OHSD datasets. The experiments primarily consist of comparative experiments and ablation studies. In the comparative experiments, we implemented four offline methods (DETR (Carion et al., 2020), Deformable DETR (Zhu et al., 2020), Mask R-CNN, and Faster R-CNN) and four online methods (Inst-GNN, ORSAD (Bresler et al., 2016), EGAT (Ye et al., 2019), and SpaceGTN). These methods were tested across multiple datasets and compared with the DiagramDiff method. It is important to note that all testing and training were conducted based on stroke datasets reconstructed from offline diagrams. For offline recognition methods, the recognized bounding boxes were used to segment the diagram into regions covered by these bounding boxes. In the ablation studies, we explored the impact of the FE module on performance and tested the recognition performance of existing methods when identifying strokes recovered from diagram images, compared to real strokes. We evaluate the performance of the model using Stroke Classification Accuracy (SCA) and Stroke Classification Precision (SCP), defined as follows:

$$SCA = \frac{\sum_{i=1}^{N} C_i}{\sum_{i=1}^{N} T_i}, \quad SCP = \sum_{i=1}^{N} \frac{T_i}{\sum_{i=1}^{N} T_i} \times P_i \qquad (11)$$

where $C_i$ represents the number of correctly classified strokes in category $i$, $T_i$ denotes the total number of strokes in category $i$, and $N$ is the total number of categories.

### 5.2.2 RESULTS AND ANALYSIS

As shown in Table 4, our method achieved state-of-the-art performance in this field. As also illustrated in Table 5, the inevitable errors in the stroke attribute information in reconstructed strokes lead to varying degrees of accuracy degradation for all methods when recognizing reconstructed strokes. Our approach leverages the FE module to effectively supplement image features with stroke attribute features rather than direct integration, significantly enhancing recognition accuracy. Using

| **Order**: Replace the email confirmation step with SMS confirmation, as SMS is more immediate. After sending the SMS, add a step to record the registration event for analytics. | **Order**: Add a step to check if the meeting room is available after checking organizer availability. If the room is not available, propose alternative times. |
|---|---|

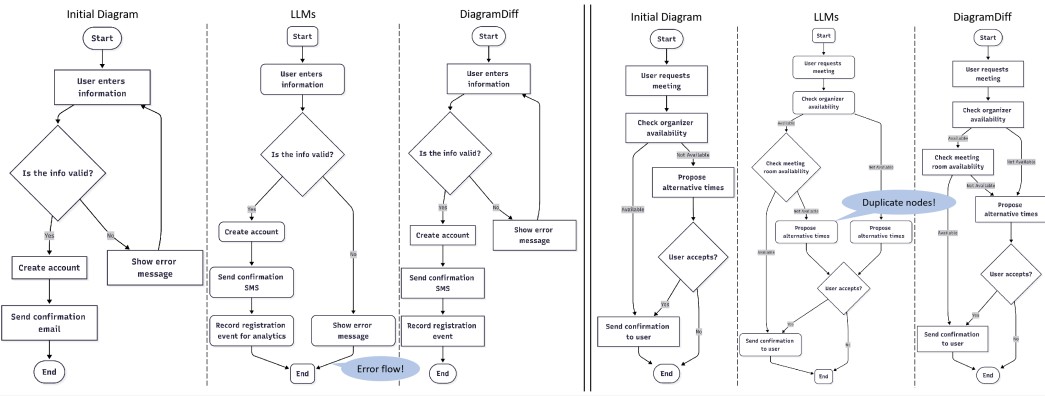

Table 6: Representative examples of diagram editing tasks in user experiments

the pre-trained diffusion model for multimodal feature fusion, our training requires only 23,134 MB of GPU memory. The average inference time per diagram is 25.2 ms, with a real-time Q&A response time of 2.7 s. The main delay comes from LLM computation, but it supports real-time interaction.

# 6  USER STUDY

Experiments are based on a dataset of 500 Q&A and editing tasks, aiming to validate the effectiveness of DiagramDiff in enhancing the diagram understanding capabilities of LLMs. The experiment uses the DiagramQAE dataset and selects LLMs including GPT-4o, Claude 3.7, DeepSeek R1 and GPT 4.5, the current advanced LLMs for diagram image understanding. Experiments divided into two groups: **Baseline Group**, where LLMs are directly used to perform the offline diagram Q&A and editing tasks, and **Comparison Group**, where DiagramDiff is applied to standardize the input offline diagram before LLMs perform the same tasks. The experiment was conducted with 10 participants, consisting of 5 males and 5 females. All participants were regular users of LLMs and had prior experience in diagram editing. Each participant randomly selected 10 diagrams from the dataset and performed 1 randomly selected Q&A task and 1 editing task for each diagram. As shown in Table 3, our method significantly outperformed the Baseline Group (direct use of LLMs) in both diagram Q&A and editing tasks. Some examples of the editing tasks performed in the user study are shown in Table 6. More examples and questionnaire outcomes can be found in Appendix D and E.

# 7  CONCLUSION

We propose DiagramDiff, an innovative framework designed to enhance the offline diagram understanding capability of LLMs. This framework overcomes the limitations of LLMs that can only address simple diagram-related questions, enabling them to support complex semantic reasoning, logical validation, and offline diagram editing. We introduce a state-of-the-art diagram stroke reconstruction method that significantly improves the reusability of offline diagrams. Additionally, we design a novel diagram recognition framework that employs a diffusion model in stroke-level diagram recognition, addressing the issue of decreased recognition accuracy in diagrams recovered from offline charts and generating standardized diagram data structures to enhance the diagram understanding capability of LLMs. Furthermore, we construct a dataset comprising 100 diagrams and 500 question-answering and editing tasks to validate the effectiveness of our method. Results demonstrate that DiagramDiff significantly enhances LLMs' Q&A and editing capabilities on complex offline diagrams

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
