# OpenReview forum: "DiagramDiff:A Diagram Reconstruction and Recognition Method to Enhance Large Language Models' Diagram Understanding"
_ICLR.cc/2026/Conference — ICLR 2026 Conference Withdrawn Submission_

### Official Review · Reviewer_NTw3 · 2025-10-27

**Soundness:** 2
**Presentation:** 1
**Contribution:** 2
**Rating:** 2
**Confidence:** 3

**Summary:**

The authors introduce DiagramDiff, a two-stage pipeline for improving LLMs’ comprehension of offline diagrams. The method first reconstructs diagram strokes from images using a segmentation and connection strategy, followed by a recognition model that combines a diffusion model and a Graph Transformer Network (GTN) to mitigate attribute noise and perform instance-level classification. The output is a structured, symbolic representation usable by LLMs for question answering (Q&A) and editing. A new benchmark dataset, DiagramQAE, is also proposed, containing 100 diagrams and 500 tasks. Experiments on several benchmarks and user studies show that DiagramDiff improves reconstruction and LLM-based diagram interaction accuracy over prior baselines.

**Strengths:**

1. **Motivated contribution to a growing multimodal area**. Diagram understanding remains underexplored in LLM research. This work meaningfully attempts to bridge diagram images and language-based reasoning systems.

2. **Strong empirical performance**. DiagramDiff outperforms prior methods in diagram stroke reconstruction and recognition, and shows clear gains in downstream editing and Q&A tasks using various LLMs.

3. **New dataset for offline diagram evaluation**. The **DiagramQAE** dataset includes diverse diagram types and task formulations, filling a gap in structured diagram evaluation for LLM.

**Weaknesses:**

1. **Method explanations are hard to understand.** Many parts of the methodology (Sec. 3) are written with document recognition experts in mind and lack clear visual or conceptual support. Some parts are really hard to understand without visual explanation. I usually consider a writing/presentation issue as a minor weakness, but this seems to be a major weakness for this submission. I highly suggest adding visual explanation about
  - 8-connected (e.g., Eq. 1)
  - input-output pairs of segment classification (Sec. 3.1.1) / stroke reconstruction (Sec. 3.1.2) / diagram recognition model (Sec. 3.2)
  - architecture of diagram recognition model (Sec. 3.2)
  - how the 'standardized diagram data structure' (Sec 3.3) looks like. What are the inputs to LLMs (with and without your methods)?
2. **Scope limitation is not clearly stated.** The method is clearly tailored to **flowchart-like diagrams** with nodes and directional edges (as seen in dataset design and supported symbols in Appendix C). However, this restriction is not acknowledged early in the paper. Many diagrams (e.g., Venn diagrams, geometry diagrams, timelines, circuit diagrams, tree map, charts) do not follow this structure and would not be well-supported by this approach.
3. **Three-sample attention via diffusion lacks justification.** The use of three independently sampled denoising results from a diffusion model as Query/Key/Value (Eq. 8–9) lacks theoretical or empirical backing. Since these are stochastically generated, their semantic meaning as Q/K/V vectors is unclear, and this design may introduce noise rather than useful diversity. This should be either better justified or replaced with a more principled feature fusion strategy.
4. **Missing or incomplete definitions in equations.** Multiple equations are introduced without complete variable definitions, reducing clarity. For example
  - what is pixel-response value in Eq. 2 and how are they obtained?
  - what is deg_M in Eq. 4?
  - which parameters in Eq. 4 learned? how are they defined?
5. **Unclear definitions of key terms like “offline” and “online” diagrams.** The distinction between “offline” and “online” diagrams is central to the paper but not clearly defined. Clarifying this early (e.g., offline = image-based, online = vector/stroke data) would help orient the reader.
6. **Only one ablation study.** Aside from the FE module, the contribution of proposed components (e.g., the split-point detection module) and other is not isolated experimentally.
7. **Efficiency comparison.** The authors mention inference time (25.2ms) and memory (23GB) but provide no comparison with baseline methods, making it hard to evaluate practicality.
8. **Qualitative examples are not well-analyzed.** Appendix E shows editing examples comparing baseline and DiagramDiff output, but these lack explanation about *why* baseline answers are incorrect and *how* the proposed method resolves them. Without interpretation, these examples have limited value to the reader.

**Questions:**

1. Is Appendix C underdeveloped? Appendix C refers to “20 categories of diagram elements” but contains only a table of names with no description, explanation, or discussion of why these categories were chosen or how robust the model is to category imbalance.

2. The citation for CASIA-OHFC seems to be missing.

---

### Official Review · Reviewer_7A8F · 2025-10-28

**Soundness:** 2
**Presentation:** 1
**Contribution:** 2
**Rating:** 2
**Confidence:** 4

**Summary:**

As offline diagrams are usually presented as unstructured image formats that hinder LLMs’ ability of diagram understanding, the authors propose DiagramDiff, a framework with a high-precision diagram reconstruction model and an instance-level diagram element recognition model. This framework enhances offline diagrams by rendering their structural information, enabling enhanced LLM-based diagram understanding. The authors also build a new dataset DiagramQAE with numerous tasks, and the experiments demonstrate the effectiveness of DiagramDiff.

**Strengths:**

1. This paper proposes a novel stroke reconstruction method and a novel diagram recognition method, making the offline diagrams structured for easy LLM-based diagram understanding.

2. This paper proposes a novel offline diagram QA and editing dataset DiagramQAE as the first offline diagram understanding dataset.

3. Experiments show the effectiveness of DiagramDiff in enhancing LLM-based diagram understanding.

**Weaknesses:**

1. The presentation is rough with many details unclear and figures and tables providing insufficient information. For example:

- The introduction needs further paragraph segmentation and explanations, as the current version are hard to comprehend.

- In figure 2(a), what is the difference between the input image and the reconstructed diagram? What are the inputs and outputs of the framework modules and how do they transform? Some modules are not described in the main text (e.g., adaptive binarization, image thinning), and what do they stand for?

- In figure 2(b), how do the outputs of the image encoder and attribute features contribute to Q and K? Where are the GCN and DWConv features? What are the difference between the output and the reconstructed diagram?

2. The methods section is disorganized and the objectives and implementation methods are not intuitive enough. For example:

- What are the targets of segments classification and stroke reconstruction? No visual samples and formulations are presented along with the unclear textual description.

- Images with dense and fuzzy crossing are eliminated due to the less readability, which is not reasonable as they are critical in some scenarios (e.g., circuit, network).

- In feature enhancement, how to obtain three intermediate sampling results? Are there any difference and specific details of the denoising process? What about GCN and DWConv features? Are they training-free or training-based methods?

3. The introduction and analysis of the experimental section are too brief, with much unclear content. For example:

- Sec.5.1.2 presents the main results without additional analysis at all. Besides, what is “FE” in table 2, 3, 5?

- Table 3 compares DiagramDiff with raw LLM without additional baselines, making the results less persuasive.

- The diagram recognition model integrates diffusion models with GCN and DWConv features, yet their contributions are not evaluated as an ablation study.

**Questions:**

See weakness part for details.

---

### Official Review · Reviewer_B95X · 2025-10-28

**Soundness:** 2
**Presentation:** 2
**Contribution:** 2
**Rating:** 4
**Confidence:** 2

**Summary:**

This paper proposes DiagramDiff, a framework aimed at enhancing large language models’ (LLMs) ability to deal with offline diagrams. The method first reconstructs diagram strokes using geometric cues such as spatial proximity, angle consistency, and curvature similarity. A diffusion model is then incorporated into a Graph Transformer Network (GTN) to improve the robustness of diagram recognition. Finally, the reconstructed and recognized elements are organized into a standardized graph structure, enabling integration with downstream LLM-based reasoning and editing tasks. In addition, the authors introduce a benchmark dataset, DiagramQAE, which includes 100 diagrams and 500 tasks for evaluating diagram understanding and question answering.Experiments show DiagramDiff achieves significantly enhancing on DiagramQAE.

**Strengths:**

- **Comprehensive technical pipeline**
  The paper presents DiagramDiff, a pipline for offline diagram reconstruction and recongnition.It introduces a stroke-reconstruction method based on spatial proximity, angular alignment, and curvature similarity, integrates a diffusion model into a GTN for diagram recognition, and designs a standardized data structure to link reconstructed diagrams with downstream LLM reasoning and editing tasks.


- **Addresses an under-explored research gap**
  The work focuses on evaluating LLMs’ understanding and editing capabilities for offline diagrams. The authors also construct the DiagramQAE dataset, consisting of 100 diagrams and 500 Q&A and editing tasks.

**Weaknesses:**

- **Insufficiently convincing motivation**
  The main contribution of the paper lies in the proposed DiagramDiff pipeline, which addresses LLM-based diagram question answering and editing tasks. However, the contribution of each module to the overall pipeline is not clearly explained and lack validation through ablation experiments.The motivation for the stroke-segmentation approach (Section 3.1.2) is demonstrated only through a hypothetical example in Fig. 3, which is too simplistic and lacks real-world representativeness. The paper claims the diffusion module reduces attribute bias in reconstructed diagrams, but offers no direct evidence.

- **Robustness lacks supporting evidence**
  The proposed reconstruction algorithm involves numerous hyperparameters—`α, β, γ, λ, µ` in Eq. (3); `θ₀, ρ, ℓ₀` in Eq. (5); and `η, τ_j` in Eq. (2)—but the paper does not specify their selection strategy and sensitivity. Particularly, the threshold parameters in Eqs. (2) and (5) have a critical impact on the effectiveness of the proposed method. This raises concerns about reproducibility and generalization.

- **Limited qualitative evaluation**
  Since the core contributions lie in the stroke-reconstruction and diagram-recognition modules, visual comparisons are critical to demonstrate superiority over baselines. However, the paper primarily reports quantitative metrics without showcasing qualitative cases.

**Questions:**

1. The authors argue that reconstruction introduces attribute bias that degrades recognition, and that the diffusion-based feature enhancement (FE) module alleviates this issue. However, Table 5 only compares existing online methods on original vs. reconstructed data, while DiagramDiff itself is evaluated only on reconstructed inputs. Could the authors provide results on original (non-reconstructed) diagrams to verify that DiagramDiff indeed improves robustness, rather than benefiting solely from adaptation to the reconstructed domain?

2. Line 182 introduces a pixel-response denoising term R(p), and its motivation and role require further explanation. how is the joint-point probability threshold  τ_j  determined?

3. The example in Fig. 3(a)(b) appears simplified and may not represent real-world diagram cases，for example will the strokes be split when online? could the authors present a real example from their dataset to illustrate stroke segmentation and reconstruction performance, possibly compared across different algorithms? how is the sliding-window size for split-point detection chosen? has its stability been tested under defferent scales?

---

### Official Review · Reviewer_8yzG · 2025-10-29

**Soundness:** 1
**Presentation:** 2
**Contribution:** 1
**Rating:** 2
**Confidence:** 4

**Summary:**

This paper proposes DiagramDiff, a framework designed to enhance large language models' (LLMs) understanding of offline diagrams by converting them into standardized and structured representations. The method combines a diagram reconstruction model (for stroke extraction and reconstruction) and a diagram recognition model that integrates a diffusion model with a graph transformer network to mitigate attribute bias. The authors also introduce DiagramQAE, a dataset of 100 diagrams paired with 500 Q&A and editing tasks, to evaluate the framework's ability to improve LLM-based reasoning and interaction with diagrams. Experiments show performance gains on reconstruction and recognition metrics compared with baselines, as well as improvements in diagram Q&A when integrated with popular LLMs.

**Strengths:**

1. **Relevant topic and motivation**: The idea of improving diagram understanding for LLMs is timely and relevant, as multimodal models still struggle with diagrammatic reasoning and structural comprehension.

2. **Reasonable conceptual direction**: Integrating structured representations before feeding visual data to LLMs could, in principle, enhance interpretability and controllability in diagram reasoning.

3. **Dataset effort**: Constructing a dataset that combines diagram reconstruction, recognition, and Q&A tasks shows initiative toward a unified benchmark, even if the scale and novelty are limited.

**Weaknesses:**

1. **Unclear motivation and background.**
The introduction lacks a clear motivation and conceptual framing of the problem. The definition of "diagram" is too general, yet the work mostly focuses on flowcharts. The paper should explicitly delimit the scope and clarify why flowcharts are chosen as representative.

2. **Unclear and limited contribution.**
It is difficult to identify a strong contribution on either the dataset or method side:

   On the dataset side, there already exist annotated flowchart benchmarks such as FlowchartQA and FlowVQA, which support question answering and structural analysis without requiring reconstruction. The novelty of DiagramQAE is therefore limited.

   On the method side, the framework appears technically redundant, combining several standard components (stroke extraction, graph-based recognition, diffusion-based fusion) without clear justification of their necessity or novelty. The formalized problem of "diagram reconstruction and recognition for LLM enhancement" lacks conceptual grounding or empirical validation.

3. **Insufficient experimental insight.**
The experiments largely report numerical improvements without qualitative analysis or causal interpretation of why the proposed components help. The link between diagram reconstruction accuracy and LLM reasoning performance remains anecdotal rather than rigorously tested.

4. **Writing and presentation issues.**
The paper contains grammatical errors and typographical mistakes (e.g., "form.such" on line 142). Figures and tables are dense, and the narrative flow between sections (especially Methods -> Experiments -> User Study) could be more coherent. The paper would benefit from substantial language editing.

**Questions:**

See weakness

---

### Note · Authors · 2025-11-12

I have read and agree with the venue's withdrawal policy on behalf of myself and my co-authors.